# Historical Test-time Prompt Tuning for Vision Foundation Models

**Jingyi Zhang**[1], **Jiaxing Huang**[1], **Xiaoqin Zhang**[2], **Ling Shao**[3], **Shijian Lu**[1]*
[1] College of Computing and Data Science, Nanyang Technological University, Singapore
[2] College of Computer Science and Technology, Zhejiang University of Technology, China
[3] UCAS-Terminus AI Lab, University of Chinese Academy of Sciences, China

## Abstract

Test-time prompt tuning, which learns prompts online with unlabelled test samples during the inference stage, has demonstrated great potential by learning effective prompts on-the-fly without requiring any task-specific annotations. However, its performance often degrades clearly along the tuning process when the prompts are continuously updated with the test data flow, and the degradation becomes more severe when the domain of test samples changes continuously. We propose HisTPT, a Historical Test-time Prompt Tuning technique that memorizes the useful knowledge of the learnt test samples and enables robust test-time prompt tuning with the memorized knowledge. HisTPT introduces three types of knowledge banks, namely, local knowledge bank, hard-sample knowledge bank, and global knowledge bank, each of which works with different mechanisms for effective knowledge memorization and test-time prompt optimization. In addition, HisTPT features an adaptive knowledge retrieval mechanism that regularizes the prediction of each test sample by adaptively retrieving the memorized knowledge. Extensive experiments show that HisTPT achieves superior prompt tuning performance consistently while handling different visual recognition tasks (e.g., image classification, semantic segmentation, and object detection) and test samples from continuously changing domains.

## 1 Introduction

Vision Foundation Models (VFMs) [1, 2, 3] have demonstrated impressive zero-shot generalization capabilities over various downstream tasks at the cost of domain expertise for crafting appropriate task-specific prompts [4, 5, 6]. To circumvent this limitation, prompt learning [4], which aims to adapt VFMs to fit downstream tasks by optimizing prompts as learnable vectors with few-shot task training samples, has been extensively explored recently. However, existing prompt tuning methods generally suffer from two constraints: 1) they require labelled training data for each downstream task which can be tedious and laborious to collect [7, 8], and 2) the learnt prompts tend to overfit to the few-shot training samples, leading to degraded generalization toward downstream tasks [9, 10, 11]. Test-time prompt tuning [7] instead learns prompts with a online flow of unlabelled test samples during the inference stage. It has attracted increasing attention recently as it allows learning effective prompts on-the-fly without requiring any task-specific annotations as illustrated in Fig. 1 (a).

Existing test-time prompt tuning methods usually start with an initial template prompt like "a photo of a [class]" and optimize it with a self-supervised objective over test images together with their model predictions [7, 8]. However, these methods often experience a clear performance degradation along the tuning process when the prompts are continuously updated with the test data flow, largely due to the lack of test-sample annotations as illustrated in Fig. 1 (b). Specifically, these methods

---

*Corresponding author

38th Conference on Neural Information Processing Systems (NeurIPS 2024).

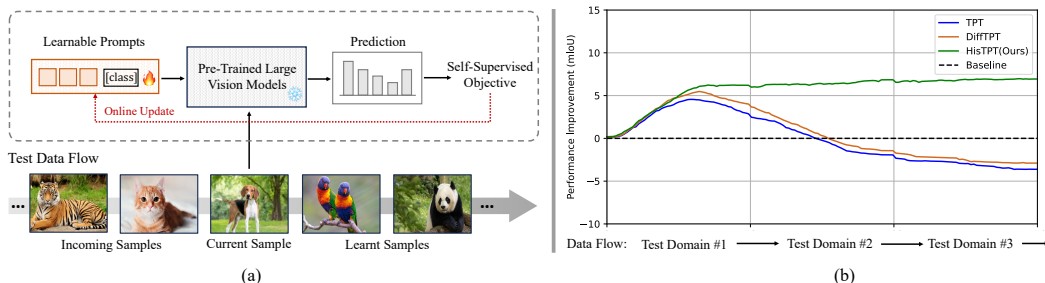

Figure 1: (a) Test-time Prompt Tuning learns and optimizes prompts from a continuous flow of unlabelled test samples during the inference stage. (b) Most existing test-time prompt tuning methods such as TPT [7] and DiffTPT [8] tend to 'forget' historical knowledge learnt from previous test samples when the prompts are continuously updated with the test data flow. They learn effective prompts at early tuning stage, but the learnt prompts degrade gradually along the tuning process. This phenomenon becomes more apparent when the domain of test samples changes continuously. The curves are derived from 100 runs over 3 different domains [16, 17]. In each run, the order of the 3 domains as well as the samples within each domain is randomly shuffled to simulate continuously changing test domains.

learn prompts well at the early test-time tuning stage, and the learnt prompt outperforms the initial template prompts clearly. However, while the tuning continues, the learnt prompts deteriorate and gradually perform even worse than the initial template prompt especially when the test domain changes continuously. These results show that existing methods [7, 8] learn effective prompts via self-supervised objectives at the early training stage, but tend to forget the useful knowledge learnt from previous test samples, and the forgetting is largely due to the accumulation of prediction errors over the unlabelled test samples along the tuning process [12, 13].

Inspired by prior studies [14, 15] in memory-based learning, we propose Historical Test-time Prompt Tuning (HisTPT) that introduces three types of knowledge banks to help memorize the previously learnt useful knowledge to mitigate the knowledge 'forgetting' problem. The three types of knowledge banks are local knowledge bank, hard-sample knowledge bank and global knowledge bank, each of which stores complementary historical information and works with different mechanisms. Specifically, local knowledge bank buffers fresh information from the recent batches of test images, capturing up-to-date distribution changes. Hard-sample knowledge bank identifies and stores the features of hard samples from local knowledge bank, capturing difficult and rare corner cases along the tuning process. Global knowledge bank stores global information by accumulating the features from the local knowledge bank and hard-sample knowledge bank, leading to comprehensive memorization that captures representative features. In addition, HisTPT introduces an adaptive knowledge retrieval mechanism which retrieves memorized knowledge adaptively for each test image for prediction regularization and prompt optimization. To this end, HisTPT builds up comprehensive memorization that preserves useful knowledge from previous test samples, mitigating the knowledge forgetting and enabling robust test-time prompt tuning as illustrated in Fig. 1 (b).

The contributions of this work can be summarized in three aspects. First, we design HisTPT, a general test-time prompt tuning framework that explores memory learning to learn effective prompts on-the-fly. To the best of our knowledge, this is the first work that explores memory learning for test-time prompt tuning. Second, HisTPT constructs three types of knowledge banks that store complementary historical information and introduces an adaptive knowledge retrieval mechanism that retrieves memorized knowledge adaptively for each test image, mitigating the 'forgetting' of learnt useful knowledge along the prompt tuning process and ultimately leading to robust prompt learning with unlabelled test samples. Third, extensive experiments over multiple benchmarks show that HisTPT achieves superior performance consistently across different visual recognition tasks such as image classification, semantic segmentation, and object detection, especially when the domain of test images continuously changes.

## 2   Related Work

**Test-time Adaptation**, which is a type of domain adaptation technique [18, 19, 20, 21], aims for designing the technique to improve model generalization over test samples [22, 23, 24]. Early studies such as test-time training (TTT) and its variants [22, 23], introduce auxiliary tasks (e.g., rotation prediction task [25]) into the supervised training objective to improve the model generalization at the training stage, and then adapt the pre-trained model to test samples via self-supervised objectives at the inference stage. Differently, recent studies [24, 20, 26, 27, 28, 29, 30, 31] generally focuses on fully test-time adaptation, where the model is adapted to test samples only during the inference stage, without introducing any auxiliary task into the training phase. For example, TENT [24] minimizes the batch-wise prediction entropy for test images while MEMO [27] enforces the prediction consistency between different augmentations of each test sample. With the advent of vision foundation models (VFMs), test-time prompt tuning [7, 8] has recently been explored for adapting pre-trained VFMs toward downstream tasks via prompt tuning at the inference stage.

**Prompt Learning of Vision Foundation Models (VFMs) [1, 2, 3]** has been studied extensively as VFMs despite their impressive zero-shot generalization capabilities over various downstream tasks often require to design appropriate task-specific prompts for optimal adaptation. Inspired by the "prompt learning" in NLP [32], one typical prompt learning approach for VFMs [4, 9, 33, 34, 35, 36, 37, 38, 39, 40, 41] learns to optimize prompts as learnable vectors with few-shot labelled samples of downstream tasks. Despite its effectiveness, it requires to label task-specific training data which is often laborious with poor scalability [7]. In addition, the learnt prompts tend to overfit to few-shot task samples, and this often degrades the generalization of VFMs while adapting toward various downstream tasks [7]. Different from prompt learning, test-time prompt tuning [7, 8] explores a new prompt learning setup that learns prompts on-the-fly with an online flow of unlabelled test images during the inference stage.

**Test-time Prompt Tuning (TPT)** aims to learn prompts on-the-fly using the test samples at inference. It has attracted increasing attention recently [7, 8, 42, 43, 44, 45] as it can learn effective prompts online with unlabelled test samples flow continuously. Most existing test-time prompt tuning studies focus on image classification tasks [7, 8, 42, 43, 44, 45]. For example, TPT [7] optimizes prompts by minimizing the prediction entropy between each test sample and its augmented views. DiffTPT [8] improves the TPT by introducing the pre-trained diffusion model [46] to produce multiple diverse and informative augmented views. Different from these studies [7, 8, 42, 43, 44, 45], HisTPT aims to mitigate the knowledge 'forgetting' problem in test-time prompt tuning when the text tokens are continuously updated with the test data flow. HisTPT achieves it by constructing comprehensive memorization capturing useful historical knowledge. In addition, HisTPT achieves superior performance across various visual recognition tasks consistently, and it can effectively handle the challenging scenario where the domain of test samples changes continuously.

**Memory-based Learning** has been studied extensively in computer vision [12, 47, 48, 49, 50, 51, 52, 53, 54, 55, 56, 57], such as semi-supervised learning [51, 58], long-term video understanding [15, 59] and domain adaptation [60, 61, 14]. For the adaptation of vision foundation models (VFMs), several studies employ memory for improving the performance on downstream tasks [62, 63, 64, 65, 66]. For instance, [66] tackles image captioning challenge by memorizing visual-related sentences which helps VFMs to generate high-quality captions with fewer hallucinations. [65] replaces text features by identity-specific sequence features extracted by CLIP, which effectively facilitates video-based person re-identification. [64] and [62] enable efficient training-free VFMs adaptation by caching category-specific data features. Different from these studies, HisTPT designs three types of knowledge banks for memorizing useful knowledge learnt from previously test samples and introduces an adaptive knowledge retrieval mechanism that retrieves memorized knowledge for each test sample adaptively, aiming for mitigating the knowledge 'forgetting' problem in test-time prompt tuning.

## 3   Method

### 3.1   Preliminaries and Task Definition

**Preliminaries of Vision Foundation Models (VFMs).** We denote a pre-trained VFM by $F = \{F^I, F^T\}$, where $F^I$ and $F^T$ are image encoder and text encoder respectively. Given a test image $x \in \mathcal{X}_{test}$ and the names of its possible belonged classes $y^c \in \mathcal{Y}_{test} = \{y^c\}_{c=1}^{C}$, the VFM image

encoder and text encoder can produce image features and category-wise text features, respectively , i.e., $v = F^I(x)$ and $u^c = F^T(y^c)$. The predictions can be obtained by calculating the similarity between the image features and the category-wise text features:

$$\hat{c} = \arg\max_c p^c, \ \ p^c = \frac{\exp\left(cos(u^c, v)\right)/\ \tau}{\sum_{j=1}^{C} \exp\left(cos(u_j, v)\right)/\ \tau}, \tag{1}$$

where $cos(\cdot)$ denotes the cosine similarity, and $\tau$ is a temperature hyper-parameter that controls the density of the encoded feature.

Instead of directly obtaining text features using the raw class names, certain hand-crafted template prompts, e.g., "a photo of a [class]", are often adopted for generating task-related textual descriptions. However, designing appropriate prompts for each downstream task is a non-trivial task which often requires domain expertise. To this end, prompt learning [4, 9] has been extensively studied, aiming to adapt VFMs to fit downstream tasks by optimizing prompts as learnable text tokens with few-shot task samples. Specifically, $M$ learnable text tokens are adopted to append the raw class names, i.e., $\mathbf{t} = \{t_1, t_2, ..., t_M\}$ each being a vector of dimension $D$ (e.g., $D = 512$). Thus, the textual description for class $c$ becomes $(\mathbf{t}; y^c)$. The learnable text tokens $\mathbf{t}$ are optimized with a task-related loss (e.g., cross-entropy loss) over the few-shot labelled training samples.

**Task Definition.** Different from conventional prompt learning, this work focuses on continual test-time prompt tuning that adapts VFMs via prompt tuning with unlabelled test images. The objective of test-time prompt tuning is to optimize the text tokens $\mathbf{t}$ for test image $x$ with certain self-supervised training losses $\mathcal{L}_{self}$ that can be formulated by:

$$\mathbf{t}* = \arg\min_{\mathbf{t}} \mathcal{L}_{self}(F, \mathbf{t}, x). \tag{2}$$

Note that the test data is presented in a continuous flow, where the text tokens are continuously updated with the test data flow.

## 3.2 Historical Test-time Prompt Tuning

We design three types of knowledge banks to help memorize the useful knowledge learnt from the previous test samples and adaptively exploit the memorized knowledge for regularizing the prediction of the current test samples. As illustrated in Fig. 2, *local knowledge bank* buffers features of the recent test images, capturing up-to-date distribution changes along the tuning process. *Hard-sample knowledge bank* actively identifies and stores hard samples from the local knowledge bank, which helps to capture difficult and corner features. *Global knowledge bank* maintains global and representative information along the whole prompt tuning process by accumulating all the features from the local knowledge bank and hard-sample knowledge bank. In addition, HisTPT introduces an *adaptive knowledge retrieval mechanism* that adaptively retrieves relevant memorized knowledge for prediction regularization and prompt optimization for each test image.

Given a continuous flow of $N$ test samples $\mathcal{X}_{test} = \{x_n\}_{n=1}^N$, we take the time step $n$ as an example to describe the knowledge bank construction with the previous test sample $x_{n-1}$ and the prompt optimization of the current sample $x_n$ with the memorized knowledge.

**Knowledge Bank Construction.** HisTPT comes with three types of knowledge banks for capturing fresh and representative knowledge during the test-time prompt tuning with previous test samples.

*Local Knowledge Bank* captures and stores fresh and up-to-date knowledge by buffering the features of the recent test samples. It works as a FIFO queue with a fixed size of $L$, where the features of the oldest test sample will be dequeued and the features of the most recent test sample will be enqueued to update the local knowledge bank, i.e, $\mathcal{M}_{local} = \{u_{local}^l, p_{local}^l\}_{l=1}^L$ on the flow. Specifically, for the latest test sample $x_{n-1}$ and its learnt text tokens $\mathbf{t}_{n-1}$, local knowledge bank enqueues its text feature $u_{n-1}$ and prediction probability $p_{n-1}$, i.e., $u_{n-1} = \{u_{n-1}^c\}_{c=1}^C$ where $u_{n-1}^c = F^T((\mathbf{t}_{n-1}; y_c))$, and $p_{n-1} = \{p_{n-1}^c\}_{c=1}^C$ where $p_{n-1}^c$ is calculated via Eq. 1. Note that the size of local knowledge bank $L$ is much smaller than the total number of test samples $N$ since local knowledge bank aims to capture fresh information and up-to-date distribution changes of test samples along the test-time prompt tuning process.

*Hard-sample Knowledge Bank* identifies hard samples from local knowledge bank for capturing difficult and corner information. We identify hard samples by those having high classification

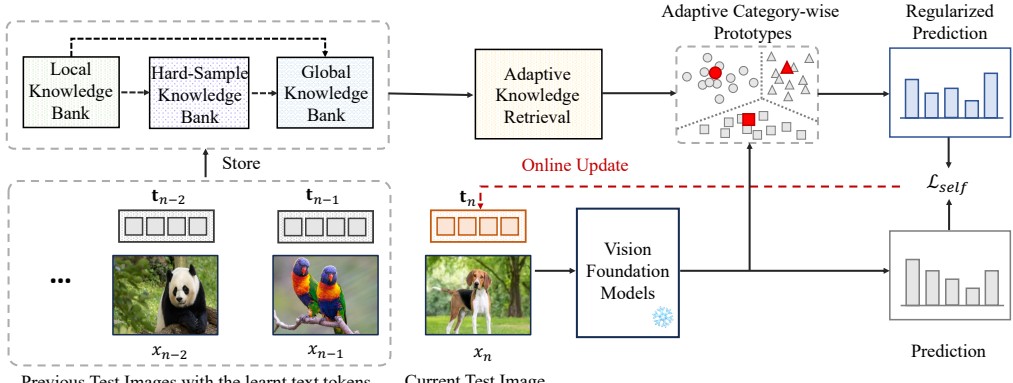

Figure 2: Overview of the proposed HisTPT. HisTPT features three types of knowledge banks, namely, local knowledge bank, hard-sample knowledge bank, and global knowledge bank, which learn and memorize up-to-date, difficult and representative knowledge, respectively, from previous test samples (e.g., $x_{n-2}$ and $x_{n-1}$) and their learnt text tokens (e.g., $\mathbf{t}_{n-2}$ and $\mathbf{t}_{n-1}$) along the test-time prompt tuning process. For the current test sample $x_n$, HisTPT regularizes its prediction by retrieving the memorized knowledge via an adaptive knowledge retrieval mechanism, enabling prompt optimization for $x_n$ with the self-supervised loss $\mathcal{L}_{self}$.

uncertainty, where the uncertainty is measured by their prediction entropy which can be computed from their prediction probability as stored in the local knowledge bank:

$$\mathcal{E}(u_{local}^l) = -\sum_{c=1}^{C} p_{local}^{(l,c)} \ \log \ p_{local}^{(l,c)}, \tag{3}$$

where the first $K$ samples with the highest entropy are selected and stored in the hard-sample knowledge bank. To enable robust memorization, we first compact the features of $K$ selected samples via category-wise average and store the compacted feature in the hard-sample knowledge bank. Similar to the local knowledge bank, hard-sample knowledge bank also works as a FIFO queue with a fixed size of $H$, i.e., $\mathcal{M}_{hard} = \{u_{hard}^h\}_{h=1}^{H}$.

*Global Knowledge Bank* stores global and representative knowledge the whole prompt tuning process by accumulating all the features from the local knowledge and hard-sample knowledge banks. Specifically, we compact the features $\bar{u}_{global}$ and $\bar{u}_{hard}$ dequeued from the local and hard-sample knowledge banks to generate category-wise feature prototype $\delta_{global} = \{\delta_{global}^c\}_{c=1}^{C}$, where $\delta_{global}^c = 1/2\,(\bar{u}_{local}^c + \bar{u}_{hard}^c)$. To facilitate stable and sustainable global memorization along the tuning process, we update the global knowledge bank with compacted feature prototype in a momentum way:

$$\delta_{global} \leftarrow (1-\gamma)\,\delta_{global} + \gamma\,\bar{\delta}_{global}, \tag{4}$$

where $\bar{\delta}_{global}$ denotes the old global feature prototype and $\gamma$ is a coefficient for smooth feature update in the global knowledge bank.

**Prompt Optimization with the Constructed Knowledge Banks.** With the built comprehensive memorization, HisTPT introduces an *Adaptive Knowledge Retrieval Mechanism* that enables adaptive retrieval of memorized knowledge for prediction regularization and prompt optimization of each test sample.

Given the test sample $x_n$ and the text tokens learnt at time step $n-1$, i.e., $\mathbf{t}_{n-1}$, the category-wise prediction probability $p_n = \{p_n^c\}_{c=1}^{C}$ can be obtained by measuring the similarity between the image feature $v_n = F^I(x_n)$ and category-wise text feature $u_n^c = F^T((\mathbf{t}_{n-1}; y_c))$ via Eq.1. The prediction $p_n$ can be enhanced via regularization with the three types of knowledge banks. For temporary knowledge in the local and hard-sample knowledge banks, we first compact the stored features into category-wise feature prototypes, i.e., $\delta_{local}$ and $\delta_{hard}$, via an average operation:

$$\delta_{local} = \{\delta_{local}^c\}_{c=1}^{C}, \delta_{hard} = \{\delta_{hard}^c\}_{c=1}^{C} \ \text{where} \ \delta_{local}^c = \frac{1}{L}\sum_{1}^{L} u_{local}^{(l,c)}, \delta_{hard}^c = \frac{1}{H}\sum_{1}^{H} u_{hard}^{(h,c)}. \tag{5}$$

The new prediction for $x_t$ can thus be obtained based on the derived prototypes $\delta_{local}$, $\delta_{hard}$, and $\delta_{global}$. Take the local prototype $\delta_{local}$ as an example. The prediction regularization of $x_n$ can be obtained with the local knowledge bank $p_{local}$ by

$$p_{local} = \{p_{local}^c\}_{c=1}^C, \ \ p_{local}^c = \frac{\exp\left(cos(\delta_{local}^c, v_n)\right)/\tau}{\sum_{j=1}^C \exp\left(cos(\delta_{local}^j, v_n)\right)/\tau}. \tag{6}$$

The prediction regularization by the hard-sample and global knowledge banks can be obtained in a similar way. Generally, the prediction with higher confidence (i.e., lower entropy) means that the corresponding feature prototype is better aligned with the current test sample in feature space, and it should contribute more to the final prediction $\hat{p}_n$ that can be obtained as follows:

$$\hat{p}_n = \sum_i w_i \, p_i, \ \ w_i = \text{Softmax}(\sum_{c=1}^C p_i^{(c)} \log \, p_i^{(c)}), \tag{7}$$

where $i \in \{local, hard, global\}$. The softmax operation is performed across the entropy of different predictions.

With the regularized prediction probability $\hat{p}_n$, the text tokens $\mathbf{t}_{n-1}$ can be optimized for the current test sample $x_n$ with the self-supervised loss defined as follows:

$$\mathcal{L}_{self} = l(p_n, \hat{p}_n) \tag{8}$$

where $l(\cdot)$ denotes a task-related loss, e.g., the standard cross-entropy loss for image classification.

## 4 Experiments

This section presents experiments including datasets, implementation details, benchmarking with the state-of-the-art, as well as discussion of our designs.

### 4.1 Datasets

We evaluate HisTPT over multiple datasets across three widely studied visual recognition tasks:

**Semantic Segmentation:** We benchmark HisTPT over 6 image segmentation datasets with pixel-wise annotations, including Cityscapes [16], BDD100K [67], Mapillary [68], ADE20K [69], Pascal Content [70] and ACDC [17].

**Image Classification:** We benchmark HisTPT over 10 classification datasets, including Flowers102 [71], DTD [72], Oxford-Pets [73], StanfordCars [74], UCF101 [75], Caltech101 [76], Food101 [77], SUN397 [78], Aircraft [79] and EuroSAT [80].

**Object Detection:** We benchmark HisTPT over 4 object detection datasets, including Cityscapes [16], BDD100K [67], ADE20K [69] and ACDC [17].

### 4.2 Implementation Details

**Semantic Segmentation:** Following [81], we adopt SEEM [3] with two vision backbones including Focal-Tiny [82] and Davit-Large [83] as the segmentation foundation models. In training, we employ AdamW optimizer [84] with a weight decay of 0.05, and set the initial learning rate as 0.0001.

**Image Classification:** Following [7, 8], we use CLIP [1] with two backbones, i.e., ResNet-50 [85] and ViT-B/16 [86], as the classification foundation models. In training, we adopt AdamW optimizer [84] with a weight decay of 0.01, and set the initial learning rate as 0.005.

**Object Detection:** For object detection task, we adopt SEEM [3] with two vision backbones including Focal-Tiny [82] and Davit-Large [83] as the detection foundation models. In training, we employ AdamW optimizer [84] with a weight decay of 0.05, and set the initial learning rate as 0.0001.

For all experiments, the prompt is initialized as "a photo of a" and the corresponding 4 tokens (i.e., $M = 4$) of dimension $D = 512$ are optimized as in [7, 8]. Unless otherwise specified, we set the size of the local knowledge bank and hard-sample knowledge bank at $L = H = 32$ and the number of the selected hard-sample features $K$ at 16. We set the update coefficient $\gamma$ of the global knowledge bank at 0.99. Following [7], we set the optimization step in test-time prompt tuning at 1 by default. All the experiments are conducted on one NVIDIA Tesla V100 GPU with batch size 1.

Table 1: Test-time prompt tuning on semantic segmentation over 6 widely adopted datasets. mIoU is reported.

| Method | Cityscapes | BDD | Mapillary | ADE | Pascal | $ACDC_{Fog}$ | $ACDC_{Night}$ | $ACDC_{Rain}$ | $ACDC_{Snow}$ | Mean |
|---|---|---|---|---|---|---|---|---|---|---|
| SEEM-Tiny | 39.2 | 37.4 | 14.7 | 14.6 | 45.1 | 34.6 | 20.7 | 33.1 | 35.8 | 30.5 |
| TPT [7] | 42.3 | 38.9 | 15.4 | 16.1 | 46.8 | 35.2 | 21.4 | 34.9 | 36.5 | 31.9 |
| TPT [7] + HisTPT | 45.1 | 41.8 | 17.5 | 17.6 | 49.4 | 37.2 | 22.9 | 37.2 | 37.8 | **34.0** |
| DiffTPT [8] | 42.9 | 39.6 | 15.8 | 16.3 | 47.1 | 35.7 | 21.6 | 35.3 | 36.6 | 32.3 |
| DiffTPT [8] + HisTPT | 45.4 | 42.1 | 16.7 | 17.9 | 49.2 | 47.6 | 22.7 | 37.7 | 38.1 | **35.2** |
| **HisTPT** | 44.7 | 41.2 | 17.2 | 17.3 | 48.7 | 36.8 | 22.1 | 36.7 | 37.1 | **33.5** |
| SEEM-Large | 49.3 | 44.6 | 18.7 | 15.2 | 37.1 | 48.1 | 32.0 | 47.4 | 45.0 | 37.4 |
| TPT [7] | 50.1 | 45.2 | 19.1 | 15.7 | 40.2 | 48.7 | 32.4 | 47.9 | 45.7 | 38.3 |
| TPT [7] + HisTPT | 52.1 | 47.4 | 21.3 | 17.1 | 45.8 | 52.1 | 33.4 | 49.4 | 48.8 | **40.8** |
| DiffTPT [8] | 50.4 | 45.7 | 19.3 | 16.1 | 41.2 | 49.1 | 32.2 | 48.2 | 46.3 | 38.7 |
| DiffTPT [8] + HisTPT | 52.4 | 47.8 | 21.1 | 17.4 | 46.3 | 52.4 | 33.6 | 49.7 | 49.1 | **41.0** |
| **HisTPT** | 51.9 | 47.3 | 20.1 | 16.9 | 45.7 | 51.6 | 33.1 | 49.1 | 48.5 | **40.4** |

Table 2: Test-time prompt tuning on image classification over 10 widely adopted datasets. Top-1 classification accuracy is reported.

| Method | Flower | DTD | Pets | Cars | UCF101 | Caltech101 | Food101 | SUN397 | Aircraft | EuroSAT | Mean |
|---|---|---|---|---|---|---|---|---|---|---|---|
| CLIP-RN50 | 61.7 | 40.3 | 83.5 | 55.7 | 58.8 | 85.8 | 73.9 | 58.8 | 15.6 | 23.6 | 55.8 |
| TPT [7] | 62.2 | 40.1 | 83.9 | 58.3 | 60.3 | 86.3 | 74.4 | 60.9 | 16.7 | 27.4 | 57.1 |
| DiffTPT [8] | 63.1 | 39.7 | 82.9 | 60.1 | 62.1 | 86.4 | 78.3 | 62.4 | 17.3 | 39.3 | 59.2 |
| **HisTPT** | 67.6 | 41.3 | 84.9 | 61.3 | 64.1 | 87.2 | 81.3 | 63.5 | 18.1 | 42.5 | **61.2** |
| CLIP-ViT-B/16 | 67.4 | 44.2 | 88.2 | 65.4 | 65.1 | 93.3 | 83.6 | 62.5 | 23.6 | 42.0 | 63.5 |
| TPT [7] | 68.2 | 47.3 | 87.1 | 66.5 | 67.7 | 93.7 | 84.2 | 65.1 | 24.3 | 42.1 | 64.6 |
| DiffTPT [8] | 69.4 | 46.3 | 87.9 | 66.4 | 68.1 | 92.3 | 86.5 | 65.3 | 25.1 | 42.8 | 65.0 |
| **HisTPT** | 71.2 | 48.9 | 89.1 | 69.2 | 70.1 | 94.5 | 89.3 | 67.2 | 26.9 | 49.7 | **67.6** |

## 4.3 Comparisons with State of the Arts

**Semantic Segmentation.** We evaluate and benchmark HisTPT over 6 semantic segmentation datasets. Since there is little prior study on test-time prompt tuning on semantic segmentation, we benchmark HisTPT by reproducing methods [7, 8], which are designed for image classification task, on semantic segmentation task. Table 1 shows experimental results. We can observe that HisTPT achieves superior segmentation performance, largely due to its comprehensive memorization that helps to regularize the predictions of test samples and mitigates the knowledge forgetting problem in test-time prompt tuning. In addition, HisTPT is complementary to existing methods and produces clear and consistent performance boosts. This is attributed to the proposed HisTPT which can effectively mitigate the knowledge forgetting existing methods.

**Image Classification.** Following [7, 8], we evaluate HisTPT over 10 image classification tasks. To suit the setup in this work, we reproduce methods [7, 8] by keeping their prompts continuously updated during the test-time adaptation. As shown in Table 2, HisTPT outperforms state-of-the-art methods consistently over different classification tasks such as classic classification on Flowers102 [71], texture classification on DTD [72] and human action recognition on UCF101 [75]. This demonstrates the superior generalization ability while HisTPT faces diverse downstream data.

**Object Detection.** We evaluate and benchmark HisTPT over 4 object detection datasets. Similar to semantic segmentation benchmarking, we benchmark HisTPT by reproducing methods [7, 8] (designed for image classification task) on the object detection task. As shown in Table 3, HisTPT achieves superior detection performance and can well handle a wide range of detection tasks including detection under various weather conditions [17] across different scenes [16, 69]. The superior detection performance is largely attributed to the knowledge banks in HisTPT which effectively help generate more accurate predictions and learn better prompts for test samples.

## 4.4 Ablation Studies

We examine the proposed HisTPT by performing ablation study over Cityscapes semantic segmentation task. As shown in Table 4, the three types of knowledge banks can work well alone and improve

Table 3: Test-time prompt tuning on object detection over 4 widely adopted datasets. $mAP_{50}$ is reported.

| Method | Cityscapes | BDD | ADE | $ACDC_{Fog}$ | $ACDC_{Night}$ | $ACDC_{Rain}$ | $ACDC_{Snow}$ | Mean |
|---|---|---|---|---|---|---|---|---|
| SEEM-Tiny | 30.5 | 26.1 | 15.7 | 44.2 | 22.3 | 25.9 | 33.9 | 28.3 |
| TPT [7] | 30.9 | 27.0 | 16.2 | 44.8 | 23.1 | 26.3 | 34.4 | 28.9 |
| DiffTPT [8] | 31.2 | 27.4 | 16.8 | 45.1 | 23.3 | 26.7 | 34.6 | 29.3 |
| **HisTPT** | 31.9 | 28.3 | 17.5 | 46.2 | 24.7 | 27.2 | 35.6 | **30.2** |
| SEEM-Large | 31.4 | 31.8 | 18.3 | 55.2 | 31.4 | 34.8 | 43.7 | 35.2 |
| TPT [7] | 31.8 | 32.2 | 18.5 | 55.6 | 31.9 | 35.1 | 44.2 | 35.6 |
| DiffTPT [8] | 32.5 | 32.3 | 18.9 | 56.1 | 32.3 | 35.4 | 44.8 | 36.0 |
| **HisTPT** | 33.2 | 33.4 | 19.4 | 56.9 | 33.1 | 36.4 | 45.2 | **36.8** |

Table 4: Ablation study of the proposed HisTPT over Cityscapes semantic segmentation task.

| Method | Histrocial Knowledge Banks | | | Adaptive knowledge retrieval | mIoU |
|---|---|---|---|---|---|
| | local knowledge bank | hard-sample knowledge bank | global knowledge bank | | |
| SEEM-Tiny | | | | | 39.2 |
| | ✓ | | | | 41.1 |
| | | ✓ | | | 40.9 |
| | | | ✓ | | 41.7 |
| | ✓ | ✓ | | | 42.2 |
| | ✓ | | ✓ | | 42.8 |
| | | ✓ | ✓ | | 42.5 |
| | ✓ | ✓ | ✓ | | 43.6 |
| **HisTPT** | ✓ | ✓ | ✓ | ✓ | **44.7** |

the performance consistently, indicating that all the stored historical knowledge is helpful in prompt tuning. In addition, the three types of knowledge banks are complementary to each other, largely because the three knowledge banks store different types of knowledge, i.e., local knowledge bank stores fresh information, hard-sample knowledge bank stores difficult corner case information, and global knowledge bank stores the global and representative features. On top of the three types of knowledge, including the proposed adaptive knowledge retrieval improves the performance further. This shows that adaptively retrieving different types of memorized information for each test image could generate more accurate prediction and ultimately lead to better test-time prompt tuning.

## 4.5 Discussion

**Complementarity to Prompt Learning Methods.** As a test-time tuning technique, the proposed HisTPT is complementary to prompt learning methods that learn prompts at the training stage. We examine this feature by setting the learnt prompts by prompt learning [4, 9] as the initial prompts of HisTPT. As Table 5 shows, equipping HisTPT with the learnt prompts improves the performance clearly, indicating that HisTPT as a plug-in can greatly enhance existing prompt learning methods.

**Optimization Steps.** We examined how the optimization step affects HisTPT by increasing it from 1 to 10. Figure 3 shows the mean mIoU over 6 semantic segmentation datasets with SEEM-Tiny. We can observe that increasing the optimization step improves segmentation consistently. Nevertheless, the performance gain becomes marginal after 6-8 optimization steps. The actual optimization step can be set by balancing the inference efficiency and the inference accuracy.

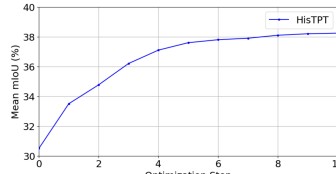

Figure 3: HisTPT with multiple optimization steps.

**Continuously Changing Test Domains.** As discussed in Section 1, HisTPT can handle challenging scenarios when the domain of test samples changes continuously. We examine this feature over semantic segmentation data that were collected under normal weather [16] and various adverse weathers [17, 87] (fog, night, rain and snow). As Table 6(a) shows, the performance of existing test-time prompt tuning methods TPT [7] and DiffTPT [8] degrades gradually along the tuning process when the weather changes from normal to adverse, largely due to increasing error accumulation and 'forgetting' while the test domain changes continuously. As a

Table 5: Complementarity to state-of-the-art prompt learning methods CoOp [4] and CoCoOp [9]. The mean top-1 accuracy across 10 image classification datasets is reported, and CoOp and CoCoOp are supervised with 16-shot labelled training data per category.

| Method | CLIP-RN50 | CoOp | CoCoOp | HisTPT | HisTPT + CoOp | HisTPT + CoCoOp |
|---|---|---|---|---|---|---|
| Mean Accuracy | 55.8 | 56.1 | 57.2 | 61.2 | 62.4 | 63.1 |

Table 6: Test-time prompt tuning on semantic segmentation across continuously changing test domains. mIoU is reported.

| Test Order (→) | Normal | Fog | Night | Rain | Snow |
|---|---|---|---|---|---|
| SEEM-Tiny | 39.2 | 34.6 | 20.7 | 33.1 | 35.8 |
| TPT | 42.3(+3.1) | 34.8(+0.2) | 20.1(-0.6) | 31.7(-1.4) | 30.6(-5.2) |
| DiffTPT | 42.9(+3.7) | 35.2(+0.6) | 20.3(-0.4) | 32.0(-1.1) | 31.4(-4.4) |
| **HisTPT** | 44.7(+5.5) | 36.9(+2.3) | 23.6(+2.9) | 37.3(+4.2) | 38.1(+2.3) |

(a)

| Test Order (→) | Snow | Rain | Night | Fog | Normal |
|---|---|---|---|---|---|
| SEEM-Tiny | 35.8 | 33.1 | 20.7 | 34.6 | 39.2 |
| TPT | 36.5(+0.7) | 34.1(+1.0) | 20.1(-0.6) | 32.7(-1.9) | 35.8(-3.4) |
| DiffTPT | 36.6(+0.8) | 34.7(+1.6) | 20.5(-0.2) | 32.9(-1.7) | 36.1(-3.1) |
| **HisTPT** | 37.1(+1.3) | 36.8(+3.7) | 22.1(+1.4) | 37.0(+2.4) | 44.9(+5.7) |

(b)

comparison, HisTPT improves the performance consistently across different weathers, and this is largely due to two factors: 1) HisTPT effectively preserves representative and up-to-date knowledge from past test samples along the tuning process; 2) HisTPT retrieves relevant memorized knowledge for each test sample, mitigating the 'forgetting' and leading to more robust test-time prompt tuning. Similar results are obtained when the test domain changes from adverse weather to normal weather as shown in Table 6(b), further verifying HisTPT's effectiveness and robustness while facing changing test domains.

**Comparisons to Existing Memory-based Learning Methods.** We examine how the proposed HisTPT performs as compared with existing memory-based learning techniques. We benchmark it with two categories of memory-based learning techniques: 1) memory-based learning in traditional network training [60, 61, 14] and 2) memory-based learning with vision foundation models [66, 65, 62]. Table 7 shows experimental results on the task of semantic segmentation on Cityscapes with SEEM-Tiny. It can be seen that HisTPT outperforms all existing memory learning techniques [60, 61, 14, 66, 65, 62] with clear margins. The superior performance is largely attributed to two factors: 1) HisTPT memorizes comprehensive knowledge of previous test samples on the fly along the prompt tuning process and 2) HisTPT features a retrieval mechanism that adaptively retrieves the memorized knowledge to learn specific prompts for each current test sample.

Table 7: Comparison with existing memory-based learning methods over Cityscapes semantic segmentation task on SEEM-Tiny. mIoU is reported.

| Method | HCL [60] | MeGA [61] | BiMem [14] | MeaCap [66] | TF-Clip [65] | TDA [62] | HisTPT |
|---|---|---|---|---|---|---|---|
| mIoU | 40.3 | 40.7 | 41.2 | 41.9 | 41.4 | 42.6 | **44.7** |

## 5 Conclusion

This paper introduces Historical Test-time Prompt Tuning (HisTPT), a general test-time prompt tuning framework that aims to mitigate the 'knowledge forgetting' problem across various visual recognition tasks. HisTPT introduces three types of knowledge banks, including local knowledge bank, hard-sample knowledge bank and global knowledge bank, each of which works with different mechanisms for memorizing useful knowledge. With the three knowledge banks, HisTPT builds up comprehensive memorization that preserves useful knowledge from previous test samples, mitigating the knowledge forgetting and enabling robust test-time prompt tuning. In addition, HisTPT comes with an adaptive knowledge retrieval mechanism that regularizes the prediction of the current test sample by adaptively retrieving the memorized knowledge. Extensive experiments show that HisTPT achieves superior performance consistently across various vision tasks. In addition, HisTPT can effectively handle the challenging scenario where the domain of test samples changes continuously. Moving forwards, we will further investigate memory-based learning for adaptation of vision foundation models.

**Acknowledgement.** This study was funded by the MOE Tier-1 project RG18/22.

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

# Appendix

## A    Datasets Details

We benchmark our HisTPT extensively over different visual recognition tasks with multiple datasets, including 10 image classification datasets, 6 semantic segmentation datasets and 4 object detection datasets. These datasets have rich diversity as shown in table 8. Specifically, the 10 image classification datasets involves a wide range of visual recognition tasks from fine-grained classification, to human action recognition and texture classification. Similarly, the images of the semantic segmentation and object detection datasets are also in rich diversity, spinning from street scene images collected from various cities with different weather conditions, to images collected under indoor scenes such as office and kitchen.

Table 8: Details of the datasets used for benchmarking HisTPT.

| Datasets | Test Images | Classes | Description |
|---|---|---|---|
| **Image Classification** | | | |
| Flower102 [71] | 2,463 | 102 | Flower images with various sizes and illumination environments. |
| DTD [72] | 1,692 | 47 | A dataset of textural images for image recognition. |
| Oxford-IIIT PETS [73] | 3,669 | 37 | A dataset for pet recognition with cat and dog images of 37 breeds. |
| Stanford Cars [74] | 8,041 | 196 | Car images for fine-grained recognition. |
| UCF101 [75] | 3,783 | 101 | A video dataset for human action recognition. |
| Caltech101 [76] | 2,465 | 101 | A dataset for common object recognition. |
| Food-101 [77] | 30,300 | 101 | Food images for fine-grained recognition. |
| SUN397 [78] | 19,850 | 397 | Indoor and outdoor scene images for fine-grained recognition. |
| Aircraft [79] | 3,333 | 100 | A dataset of 100 aircraft model variants for aircraft model recognition. |
| EuroSAT [80] | 8,100 | 10 | A dataset of satellite images for land use and land cover recognition. |
| **Semantic Segmentation** | | | |
| Cityscapes [16] | 500 | 19 | Scene images collected in different cities for street scene understanding. |
| BDD100K [67] | 1,000 | 19 | Street scene images collected at different times of the day. |
| Mapillary [68] | 2,000 | 65 | A dataset of street-level images with high resolution. |
| ADE20K [69] | 2,000 | 150 | A large-scale dataset of images collected from outdoor and indoor scenes. |
| Pascal Content [70] | 5101 | 59 | An extension of PASCAL VOC 2010 dataset with pixel-wise annotations. |
| ACDC [17] | 406 | 19 | Scene images with adverse weather conditions, i.e., fog, night, rain, snow. |
| **Object Detection** | | | |
| Cityscapes [16] | 500 | 8 | Scene images collected in different cities for street scene understanding. |
| BDD100K [67] | 1,000 | 8 | Street scene images collected at different times of the day. |
| ADE20K [69] | 2,000 | 100 | A large-scale dataset of images collected from outdoor and indoor scenes. |
| ACDC [17] | 406 | 8 | Scene images with adverse weather conditions, i.e., fog, night, rain, snow. |

## B    Parameter Analysis

We study the size of the local knowledge bank and hard-sample knowledge bank ($L$ and $H$), the parameter $K$ used in hard-sample knowledge bank update, and the update coefficient $\gamma$ used in Eq. 4 for global knowledge bank, over the semantic segmentation task Cityscapes with SEEM-Tiny.

**Size of the local knowledge bank $L$.** As discussed in the main text, the size of local knowledge bank $L$ is much smaller than the total number of test samples, since local knowledge bank aims to buffer fresh information from recent previous test samples. Here we study how it affects the test-time prompt tuning. As shown in Table 9 (a), HisTPT yields robust performance when $L$ is relatively small (from 8 to 64), while the performance drops slightly when it becomes too large. This show that the local knowledge bank with relatively small size could effectively capture fresh information and up-to-date distribution changes along the tuning process.

**Size of the hard-sample knowledge bank $H$.** Hard-sample knowledge bank stores the features of hard-samples, capturing different and rare corner cases during the test-time prompt tuning process. Table 9 (b) show that HisTPT is quite robust when $H$ is between 8 to 128. Hence, we simply set it as the same as the size of the local knowledge bank, i.e., $H = L = 32$.

**The number of selected hard-sample features $K$.** As discussed in the main text, hard-sample identifies and stores $K$ hard-sample features from the local knowledge bank. Here we study the sensitivity of $K$ by increasing it from 8 to 24 with a step of 4. As shown in Table 9(c), the performance is quite tolerant to the parameter $N$ and the best performance is obtained when $K = 16$.

**Update coefficient $\gamma$.** The update coefficient $\gamma$ in Eq. 4 determines the update speed of global knowledge bank, where the larger update coefficient results in the slower update of global knowledge bank. From Table 9 (d), we can observe that HisTPT is robust when $\gamma$ is large enough (i.e., from 0.9 to 0.999) while the performance of HisTPT drops slightly when $\gamma$ becomes too small. This demonstrates that a large update coefficient, ensuring

smooth and gradual updates, facilitates stable global memorization. Conversely, a too small update coefficient leads to rapid updates of the global knowledge bank, resulting in unstable memorization and less effective test-time prompt tuning.

Table 9: Parameter analysis of HisTPT over semantic segmentation task Cityscapes with SEEM-Tiny.

| $L$ | 8 | 16 | 32 | 64 | 128 | 512 |
|---|---|---|---|---|---|---|
| **HisTPT** | 44.5 | 44.7 | **44.7** | 44.6 | 44.2 | 43.9 |

(a) The size of local knowledge bank $L$.

| $H$ | 8 | 16 | 32 | 64 | 128 | 512 |
|---|---|---|---|---|---|---|
| **HisTPT** | 44.7 | 44.6 | **44.7** | 44.5 | 44.6 | 43.5 |

(b) The size of hard-sample knowledge bank $H$.

| $K$ | 8 | 12 | 16 | 20 | 24 |
|---|---|---|---|---|---|
| **HisTPT** | 44.6 | 44.5 | **44.7** | 44.6 | 44.6 |

(c) The number of hard-sample features $K$.

| $\gamma$ | 0.1 | 0.5 | 0.9 | 0.99 | 0.999 |
|---|---|---|---|---|---|
| **HisTPT** | 43.1 | 43.9 | 44.5 | **44.7** | 44.6 |

(d) The update coefficient $\gamma$.

## C   More Discussion about the Design of Historical Knowledge Banks

**Update of the hard-sample knowledge bank.** As discussed in the main text, hard-sample knowledge bank works as an FIFO queue with a fixed size, and it is updated using the hard-sample features selected from local knowledge bank with an average compaction operation. Here we provide more discussion about the different update ways of hard-sample knowledge bank, including 1) directly update using the selected features and 2) update using the compacted features with an average operation. From Table 10, we can observe that updating hard-sample knowledge bank using the selected features with average compaction operation performs better, which is largely due to that the compacted features enabling to filter out some noises and results in more robust memorization of difficult and corner-case information.

Table 10: Comparison of different update ways of hard-sample knowledge bank over semantic segmentation task Cityscapes with SEEM-Tiny.

| Method | Directly update | Update with average operation |
|---|---|---|
| mIoU | 43.9 | **44.7** |

**Update of the global knowledge bank.** As described in the main text, we update the global knowledge bank using the features dequeued from both the local knowledge bank and hard-sample knowledge bank. Here we study its effectiveness with different update ways of global knowledge bank, including 1) update global knowledge bank with only the features dequeued from local knowledge bank; 2) update global knowledge bank with only the features dequeued from hard-sample knowledge bank and 3) update global knowledge bank with the features dequeued from the local knowledge bank and hard-sample knowledge bank. Table 11 shows the experimental results. It can be observed that updating global knowledge bank with the features dequeued from both the local knowledge bank and hard-sample knowledge bank performs the best, which indicates that the features stored in local knowledge bank and hard-sample knowledge bank are complementary to each other, working together to help build a more comprehensive and representative global memorization.

Table 11: Comparison of different update ways of global knowledge bank over semantic segmentation task Cityscapes with SEEM-Tiny.

| Method | local knowledge bank | hard-sample knowledge bank | global& hard-sample knowledge banks |
|---|---|---|---|
| mIoU | 44.2 | 43.8 | **44.7** |

## D   More Comparisons with Memory-based Learning Methods

We provide more comparisons with existing memory-based learning methods [31, 88, 89, 28]. Our HisTPT differs in two major aspects: Memory Types - HisTPT designs three types of knowledge banks for capturing and storing both fresh and representative features; Memory Retrieval - HisTPT designs an Adaptive Knowledge Retrieval Mechanism for retrieving the memorized information adaptively for each test image. Due to the very different designs, HisTPT outperforms [31, 88, 89, 28] clearly as shown in Table 12.

Table 12: Comparison with existing memory-based learning methods over Cityscapes semantic segmentation task on SEEM-Tiny. mIoU is reported.

| Method | T3A [31] | TAST [88] | RoTTA [89] | FAU [28] | HisTPT |
|--------|----------|-----------|------------|----------|--------|
| mIoU   | 41.8     | 42.0      | 41.9       | 42.2     | **44.7** |

## E   Pseudo Codes of HisTPT

We provide the pseudo codes of the proposed historical test-time prompt tuning (HisTPT), as shown in Algorithm 1. We initialize the three knowledge banks with the features of the first test sample and then gradually update them as in Lines 3-7 along the test-time prompt tuning process. Note that, for the first test sample, we skip the prediction regularization in Line 10 and optimize the tokens for it with the vanilla self-training objective since the knowledge banks have not been constructed at that time.

---
**Algorithm 1** Historical Test-Time Prompt Tuning.

---
**Require:** Online optimized text tokens $\mathbf{t}$, a pre-trained vision foundation model $F = \{F^I, F^T\}$, a continuous flow of test samples $\mathcal{X}_{test} = \{x_n\}_{n=1}^N$ and their possible belonged class names $\mathcal{Y}_{test} = \{y^c\}_{c=1}^C$

1: **Initialization:** Initialize $\mathbf{t}$ as $\mathbf{t}_0$
2: **for** $n = 1$ **to** $N$ **do**
3:     **Knowledge bank construction with** $x_{n-1}$ **and** $\mathbf{t}_{n-1}$:
4:     Encode $x_{n-1}$: $u_{n-1} = F^T(\mathbf{t}_{n-1}; \mathcal{Y}_{test})$
5:     Update *local knowledge bank*: dequeue old feature $\bar{u}_{local}$ and enqueue $u_{n-1}$
6:     Update *hard-sample knowledge bank*: dequeue old feature $\bar{u}_{hard}$ and enqueue new feature selected by Eq. 3
7:     Update *global knowledge bank*: generate new category-wise feature prototype using $\bar{u}_{local}$ and $\bar{u}_{hard}$, and update the global knowledge bank by Eq. 4
8:     **Prompt optimization for** $x_n$ **with the constructed knowledge banks**:
9:     Generate prediction $p_n$ for $x_n$ with $\mathbf{t}_{n-1}$ via Eq. 1
10:     Generate the regularized prediction $\hat{p}_n$ by adaptively retrieving the memorized knowledge as in Eqs. 5-7
11:     Optimize the text token for $x_n$, i.e., $\mathbf{t}_n \leftarrow \mathbf{t}_{n-1}$, by Eq. 8
12: **end for**

---

## F   Quantification of the Forgetting Mitigation Ability of HisTPT

Following prior study [29], we measure the forgetting by randomly selecting one of the five datasets in Table 6 as the reference domain and perform continual adaptation toward the other four datasets. During the continuous adaptation process, we evaluate HisTPT's ability of preserving the knowledge of vision foundation models by measuring its performance on the reference domain. As shown in the Figure 4, HisTPT shows less performance degradation on the reference domain consistently, demonstrating its effectiveness in preserving the knowledge of vision foundation models and mitigating forgetting during the adaptation process.

## G   Further Analysis of the Three Knowledge Banks

We analyse the three knowledge banks by visualizing their stored features along the test-time adaptation process. Three points can be drawn as illustrated in Figure 5: 1) the global prototypes exhibit slow and gradual shift from the initial feature prototypes, preserving the knowledge of pre-trained vision foundation models and facilitating stable test-time adaptation; 2) the features in the local knowledge bank change rapidly, validating their effectiveness in capturing fresh and up-to-date distribution changes along the test-time adaptation process; 3) most features in the hard-sample knowledge bank lies around inter-category boundary, indicating their effectiveness in capturing difficult and rare corner cases along the tuning process. With the three types of complementary knowledge, HisTPT enables adaptive regularization for the prediction of current test samples.

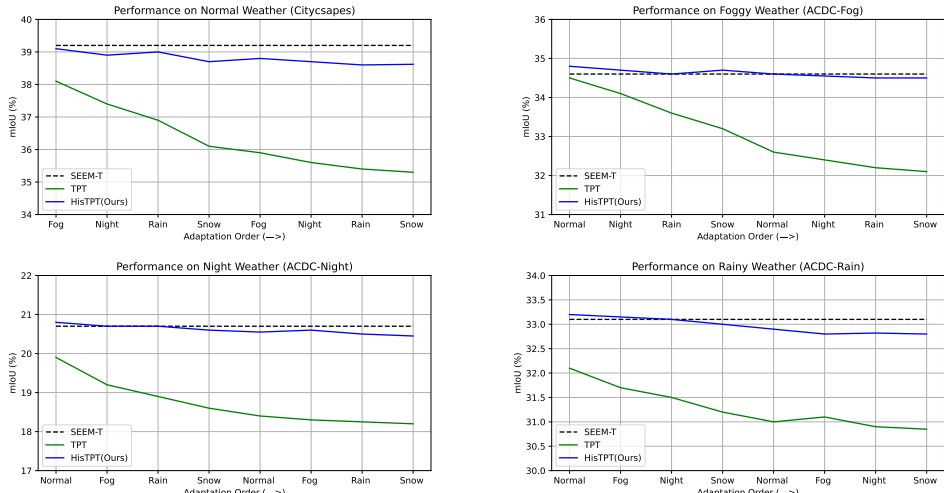

Figure 4: Comparison of preventing forgetting on continual test-time adaptation task with SEEM-Tiny. For each experiment, one dataset is selected as the reference domain, and then we perform the continual adaptation on the other datasets. We record the performance change on the reference domain for measuring the forgetting during the continual adaptation process. Our HisTPT shows clearly less performance degradation on the reference domain, demonstrating the effectiveness of HisTPT in mitigating forgetting during the adaptation process.

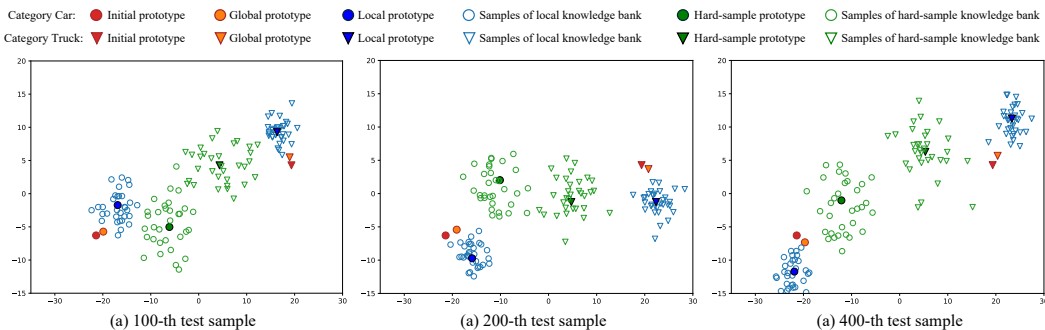

Figure 5: T-SNE visualization of the features stored in each knowledge bank with Cityscapes semantic segmentation task on SEEM-Tiny. For clear illustration, we select two categories (i.e., car and truck) for visualization. T-SNE visualization shows that 1) global prototype shifts slowly from the initial prototype, preserving the original knowledge of pre-trained vision foundation models; 2) local knowledge bank updates rapidly, capturing fresh information and reflecting real-time distribution changes and 3) hard-sample knowledge bank captures challenging and rare cases situated near decision boundaries.

# H   Analysis with Error Bars

In experiments, we observe negligible variance on the results between multiple random runs. Nevertheless, we provide the error bar with 5 random runs to analyze the proposed HisTPT on semantic segmentation task with SEEM-Tiny, image classification task with CLIP-RN50 and object detection with SEEM-Tiny, respectively. From Table 13, we can observe that our proposed HisTPT performs well consistently over multiple random runs.

Table 13: Analysis of our proposed HisTPT with error bars.

| Method | Semantic segmentation task (Mean) | Image classification task (Mean) | Object detection task (Mean) |
|--------|-----------------------------------|----------------------------------|------------------------------|
| HisTPT | 33.5 ±0.2 | 61.2 ±0.1 | 30.2 ±0.2 |

# I  Qualitative Results

We present qualitative illustrations and comparisons over semantic segmentation task on Cityscapes. As shown in Fig. 6, HisTPT yields the best segmentation consistently which is well aligned with the quantitative results.

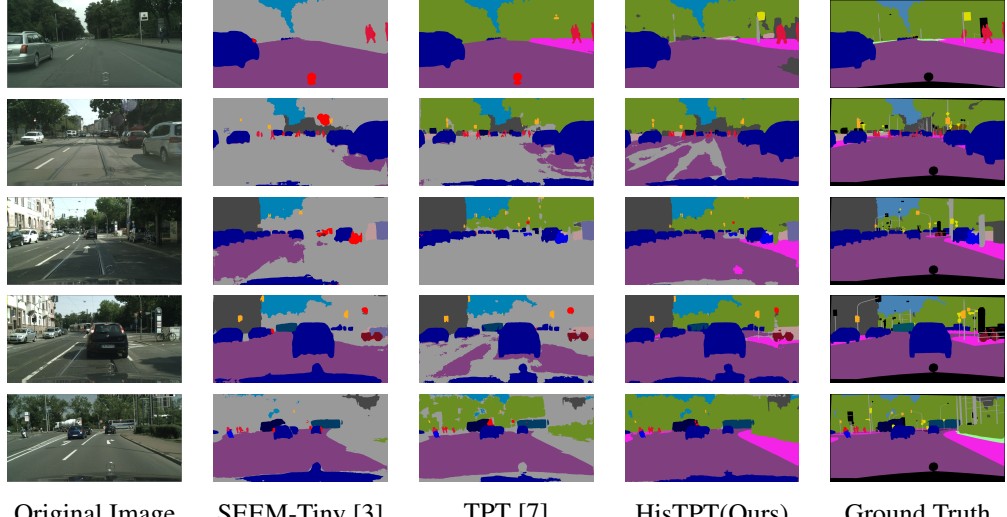

|  |  |  |  |  |
|---|---|---|---|---|
| Original Image | SEEM-Tiny [3] | TPT [7] | HisTPT(Ours) | Ground Truth |

Figure 6: Qualitative comparison of HisTPT with the baseline model (SEEM-Tiny) [3] and TPT [7] over semantic segmentation task on Cityscapes.

# J  Broader Impacts and Limitations

**Broader Impacts.** This work explores a novel pipeline for transfer learning with vision foundation models, namely, test-time prompt tuning. Our proposed method offers great advantages by eliminating the need for labelled task-specific data and allowing learning prompts from test samples on-the-fly. It thus makes a very valuable contribution to the computer vision research community by providing a novel and efficient transfer learning pipeline. The feature of requiring no labelled task-specific training data enables efficient adoption of vision foundation models in various downstream tasks, broadening the applicability of vision foundation models significantly.

**Limitations.** As discussed in Section 4.2 of the main text, HisTPT offers a general framework that can perform well across different computer vision tasks. It enables effective test-time prompt tuning with the generic text prompt that is universally applicable across all vision foundation models (VFMs), thus avoiding the complexity of task-specific designs in VFM adaptation. At the other end, task-specific designs allow incorporating task-relevant knowledge which often helps improve performance. For instance, the incorporation of specific visual prompts, such as points and bounding boxes, in segmentation or detection foundation models often lead to more precise segmentation masks and bounding boxes. We will investigate how to incorporate task-specific prompt tuning in our future work.

