# OpenReview forum: "Historical Test-time Prompt Tuning for Vision Foundation Models"
_NeurIPS.cc/2024/Conference — NeurIPS 2024 poster_

### Official Review · Reviewer_CdiG · 2024-07-08

**Soundness:** 3
**Presentation:** 1
**Contribution:** 2
**Rating:** 5
**Confidence:** 4

**Summary:**

This paper introduces a framework designed to mitigate the knowledge forgetting problem in test-time prompt tuning. The proposed method, HisTPT, employs three types of knowledge banks-local, hard-sample, and global-to memorize useful knowledge from previous test samples, thereby enhancing prompt optimization during inference. An adaptive knowledge retrieval mechanism is integrated to regularize predictions and optimize prompts based on memorized knowledge. Extensive experiments across various visual recognition tasks, such as image classification, semantic segmentation, and object detection, demonstrate HisTPT's superior performance, particularly in scenarios where test domains change continuously.

**Strengths:**

1. The introduction of knowledge banks and an adaptive knowledge retrieval mechanism provides a clear and effective solution to address the issue of knowledge forgetting during test-time prompt tuning.
2. The method is validated through extensive experiments on multiple benchmarks, showing its effectiveness across different visual recognition tasks and varying test domains.

**Weaknesses:**

1. While the three memory banks are highlighted as a solution to the knowledge forgetting issue, the paper lacks an explanation of why and how these banks effectively address this problem, relying mostly on empirical results without sufficient analysis.
2. The paper suffers from repetitive text, which affects readability and clarity. The authors often reiterate the same points, and the figures do not effectively illustrate the proposed method.

**Questions:**

N/A

---

> ### Author Rebuttal · Authors · 2024-08-06
>
> **[Response 1] Further explanation of how and why the proposed knowledge memory banks effectively address the knowledge forgetting issue:**
>
> Thank you for pointing out this issue.
>
> As discussed in Lines 37-45, HisTPT introduces three types of knowledge banks to memorize the previously learnt knowledge which helps mitigate the 'forgetting' issue. Specifically, the forgetting is largely caused by the accumulation of prediction errors over unlabelled test samples along the tuning process. HisTPT exploits the three types of complementary knowledge that collaborate to denoise the prediction of test samples along the tuning process, alleviating error accumulation and ultimately mitigating the forgetting issue.
>
> We analyze how the three memory banks help mitigate the forgetting by visualizing the change of their stored features along the test-time adaptation process. As shown in **Figure 2** of the attached PDF, the three types of knowledge banks store complementary historical features:
> 1) The global prototypes exhibit slow and gradual shift from the initial feature prototypes, preserving the knowledge of pre-trained vision foundation models and facilitating stable test-time adaptation.
> 2) The features in the local knowledge bank change rapidly, validating their effectiveness in capturing fresh and up-to-date distribution changes along the test-time adaptation process.
> 3) Most features in the hard-sample knowledge bank lies around inter-category boundary, indicating their effectiveness in capturing difficult and rare corner cases along the tuning process.
>
> In this way, HisTPT can leverage the comprehensive and complementary knowledge stored in the three memory banks to denoise predictions of test samples along the tuning process, reducing error accumulation and effectively mitigating the issue of forgetting.
>
> In addition, we follow prior test-time adaptation study [a] and conduct experiments to analyze the forgetting mitigation ability of HisTPT. Specifically, we randomly select one of the five datasets in Table 6 as the reference domain and perform continual adaptation toward the other four datasets. During the continuous adaptation process, we evaluate HisTPT's ability of preserving the knowledge of vision foundation models by measuring its performance on the reference domain. As shown in **Figure 1** of the attached PDF, HisTPT shows clearly less performance degradation on the reference domain consistently, demonstrating its effectiveness in mitigating forgetting during the adaptation process.
>
> We will include above analysis in our revised manuscript.
>
> [a] Efficient test-time model adaptation without forgetting, ICML 2022.
>
> **[Response 2] The texts and figures need to be improved:**
>
> Thank you for your comments! We will check through the manuscript carefully to remove repetitive text as suggested. In addition, we will improve Figure 2 of the main manuscript to illustrate the framework of the proposed method more clearly.

---

> ### Author Response · Authors · 2024-08-13
>
> Dear Reviewer CdiG,
>
> Thank you for your insightful feedback. We have carefully considered your questions and suggestions and have addressed them accordingly. We sincerely appreciate your constructive comments, which have helped strengthen our paper. As the discussion phase is nearing its conclusion, we would appreciate it if you could let us know if there are any additional questions or suggestions.
>
> Best regards,
>
> Authors

---

### Official Review · Reviewer_ejR5 · 2024-07-11

**Soundness:** 3
**Presentation:** 3
**Contribution:** 3
**Rating:** 5
**Confidence:** 4

**Summary:**

This paper takes a deep investigation into the memory bank and proposes a framework with three different memory banks for storing local knowledge, hard-sample knowledge, and global knowledge. The experimental results show that the proposed method delivers superior performance on many tasks.

**Strengths:**

1. This paper explore an interesting about the memory bank for test-time prompt tuning. This is direction is meaningful for making the test-time prompt tuning more robust in practice.
2. The experiments in this paper is comprehensive and solid. The results on semantic segmentation, object detection, and image classification tasks reveal the superiority of the proposed method.

**Weaknesses:**

1. Further analysis of the three memory banks should be presented to show which samples and representations are stored in the memory during the testing. This would be helpful to evaluate how each memory can help the final performance in various aspects.
2. The comparison methods are limited in this paper, e.g., [1, 2]. Please discuss them in the related work.
3. The running time of the proposed method should be reported.

[1] SwapPrompt: Test-Time Prompt Adaptation for Vision-Language Models. NeurIPS 2023

[2] Efficient Test-Time Adaptation of Vision-Language Models. CVPR 2024

**Questions:**

Please refer to the `Weaknesses` section.

**Limitations:**

The authors have discussed the limitations in the appendix.

---

> ### Author Rebuttal · Authors · 2024-08-06
>
> **[Response 1] Further analysis of the three knowledge banks:**
>
> Thank you for your suggestion! We analyze the three knowledge banks by visualizing their stored features along the test-time adaptation process. Three points can be drawn as illustrated in **Figure 2** in the attached PDF:
> 1) The global prototypes exhibit slow and gradual shift from the initial feature prototypes, preserving the knowledge of pre-trained vision foundation models and facilitating stable test-time adaptation.
> 2) The features in the local knowledge bank change rapidly, validating their effectiveness in capturing fresh and up-to-date distribution changes along the test-time adaptation process.
> 3) Most features in the hard-sample knowledge bank lies around inter-category boundary, indicating their effectiveness in capturing difficult and corner cases along the tuning process.
>
> With the three types of complementary knowledge, HisTPT enables adaptive regularization for the prediction of current test samples. We will include the above analysis and the attached visualization in the updated manuscript.
>
> **[Response 2] Comparisons with more related works:**
>
> Thank you for sharing the two prior studies! We will discuss and compare with them in the updated manuscript. Specifically, [1] leverages self-supervised contrastive learning to facilitate the test-time prompt adaptation, while [2] introduces a training-free dynamic adapter that caches category-specific feature for efficient test-time adaptation. Differently, HisTPT focuses on mitigating the knowledge 'forgetting' problem in test-time prompt tuning, and it achieves it by constructing comprehensive memorization that captures useful historical knowledge. As shown in Table below, HisTPT performs clearly better than [1,2], demonstrating its effectiveness in tackling test-time prompt tuning challenge. The experiments are conducted on Cityscapes semantic segmentation task with SEEM-Tiny.
>
> |Cityscapes|SwapPrompt[1] | TDA[2] | **HisTPT**|
> |:---:|:---:|:---:|:---:|
> |mIoU|43.4 | 43.7  | **44.7**|
>
>
> **[Response 3] Running time of HisTPT:**
>
> Thank you for your suggestion! The table below shows the run time of the proposed HisTPT with CLIP ResNet-50 on a single GPU over dataset Flowers-102. We can see that HisTPT is clearly more efficient than TPT. The better efficiency is largely attributed to two factors: 1) TPT involves heavy augmentations of test images (e.g., 64 augmentations for each test image), while HisTPT does not; 2) HisTPT introduces memory to store historical information efficiently as described in Lines 155-157, 169-172 and 541-550.
>
> |Methods|Run time per image (s) |
> |:---:|:---:|
> |TPT | 0.66|
> |HisTPT| 0.12|

---

> ### Author Response · Authors · 2024-08-13
>
> Dear Reviewer ejR5,
>
> Thank you for your insightful feedback. We have carefully considered your questions and suggestions and have addressed them accordingly. We sincerely appreciate your constructive comments, which have helped strengthen our paper. As the discussion phase is nearing its conclusion, we would appreciate it if you could let us know if there are any additional questions or suggestions.
>
> Best regards,
>
> Authors

---

> > ### Comment · Reviewer_ejR5 · 2024-08-13
> >
> > Thank you for your response. I would like to keep my positive score for this paper.

---

### Official Review · Reviewer_hT3z · 2024-07-12

**Soundness:** 3
**Presentation:** 2
**Contribution:** 3
**Rating:** 5
**Confidence:** 5

**Summary:**

The paper makes a comprehensive investigation of the memory bank in test-time prompt tuning for CLIP. To address the forgetting issue, the author proposes HisTPT. HisTPT aims to address this by memorizing useful knowledge from learned test samples, using three types of memory banks: local, hard-sample, and global. Extensive experiments on segmentation and classification demonstrate the superiority of HisTPT.

**Strengths:**

1. HisTPT is easy to follow. The proposed framework is well-introduced and the construction of memory banks is intuitive。
2. The evaluation is comprehensive. HisTPT is evaluated across multiple visual recognition tasks including image classification, semantic segmentation, and object detection, demonstrating consistent performance improvements over state-of-the-art test-time prompt tuning methods.

**Weaknesses:**

1. The main experiment does not match the motivation. The motivation emphasizes solving continuously changing test domains [1]. However, the main experiments only focus on single datasets, lacking evaluation of the continuous test-time adaptation. Alternatively, the author should make a formal claim indicating whether HisTPT addresses covariate shift or other issues.
2. The experiment is insufficient. Memory bank is a common technique in TTA [2-5]. However, the author does not discuss it in the related work, or make comparison in the experiment.

[1] Wang Q, Fink O, Van Gool L, et al. Continual test-time domain adaptation. In CVPR. 2022.

[2] Iwasawa Y, Matsuo Y. Test-time classifier adjustment module for model-agnostic domain generalization. In NeurIPS. 2021

[3] Jang M, Chung S Y, Chung H W. Test-time adaptation via self-training with nearest neighbor information. In ICLR. 2023.

[4] Yuan L, Xie B, Li S. Robust test-time adaptation in dynamic scenarios. In CVPR. 2023.

[5] Wang S, Zhang D, Yan Z, et al. Feature alignment and uniformity for test time adaptation. IN CVPR. 2023.

**Questions:**

1. The authors emphasize mitigating knowledge forgetting. However, they do not clearly define how forgetting is quantified. In TTA, performance on the original domain is typically used to measure forgetting. I recommend the authors modify their motivation or conduct experiments to demonstrate the superiority of HisTPT in mitigating the forgetting issue.

[1] Niu S, Wu J, Zhang Y, et al. Efficient test-time model adaptation without forgetting. In ICML. 2022

---

> ### Author Rebuttal · Authors · 2024-08-06
>
> **[Response 1] Evaluation of HisTPT over the continuous test-time adaptation task:**
>
> We would clarify that we evaluated HisTPT over continuously changing test domains in Table 6 of the main manuscript. As discussed in Lines 280-293, HisTPT can tackle challenging scenarios when the domain of test samples changes continuously.
>
> **[Response 2] Comparisons with other memory-based TTA methods:**
>
> Thank you for sharing the prior studies [2-5] and we will review and compare with them in the revised paper. Our HisTPT differs in two major aspects:
> 1) Memory Types - HisTPT designs three types of knowledge banks for capturing and storing both fresh and representative features;
> 2) Memory Retrieval - HisTPT designs an Adaptive Knowledge Retrieval Mechanism for retrieving the memorized information adaptively for each test image.
>
> As a comparison, the shared studies generally employ a single type of vanilla memory that stores either samples or class centroids. Due to the very different designs, HisTPT outperforms the shared work clearly as shown in the table below (on Cityscapes semantic segmentation task with SEEM-Tiny).
>
> |Cityscapes|T3A [2] | TAST [3] | RoTTA [4] | FAU [5] | **HisTPT**|
> |:---:|:---:|:---:|:---:|:---:| :---:|
> |mIoU|41.8 | 42.0 | 41.9 | 42.2 | **44.7**|
>
> **[Response 3] Quantification of the forgetting mitigation ability of HisTPT:**
>
> Thank you for your suggestion! Since the source domain of vision foundation models (pre-training) is generally huge with data from multiple sources, it is very challenging to measure the forgetting by directly testing the performance over such source domain. Instead, we randomly select one of the five datasets in Table 6 as the reference domain and perform continual adaptation toward the other four datasets. During the continuous adaptation process, we evaluate HisTPT's ability of preserving the knowledge of vision foundation models by measuring its performance on the reference domain. As shown in the **Figure 1** in the attached PDF, HisTPT shows less performance degradation on the reference domain consistently, demonstrating its effectiveness in preserving the knowledge of vision foundation models and mitigating forgetting during the adaptation process.

---

> > ### Comment · Reviewer_hT3z · 2024-08-11
> >
> > Dear Authors,
> >
> > Thank you for your response. I maintain my positive rating.
> >
> > Best,
> >
> > Reviewer hT3z

---

> > > ### Author Response · Authors · 2024-08-13
> > >
> > > Dear Reviewer hT3z,
> > >
> > > Thank you for your positive evaluation of our work. We sincerely appreciate your feedback and suggestions.
> > >
> > > Best regards,
> > >
> > > Authors

---

### Official Review · Reviewer_43p7 · 2024-07-13

**Soundness:** 2
**Presentation:** 3
**Contribution:** 2
**Rating:** 4
**Confidence:** 4

**Summary:**

This paper proposes Historical Test-time Prompt Tuning (HisTPT), aimed at addressing the performance degradation issue of test-time prompt tuning methods in scenarios where test samples continuously change. HisTPT establishes a local knowledge bank, hard-sample knowledge bank, and global knowledge bank, each storing the recent, hard, and global features of test samples, respectively. Furthermore, it employs an adaptive knowledge retrieval mechanism to compute pseudo-labels for individual test samples and carry out prompt optimization. Overall, HisTPT is not entirely new, and has achieved performance surpassing previous methods in classification, segmentation, and detection tasks. Comprehensive ablation experiments have also verified the wide applicability of HisTPT.

**Strengths:**

- HisTPT is simple yet effective, demonstrating scalability in classification, segmentation, and detection tasks.
- Sufficient experiments and ablation studies demonstrate the effectiveness of the proposed method.
- The paper is well-organized and easy to follow.

**Weaknesses:**

- Entropy is not a good metric for calibrating confidence. However, HisTPT extensively uses entropy as a tool for weight calculation. It's advised to test different confidence metrics to verify the robustness of HisTPT.
- It is unclear how the ablation experiments are designed without Adaptive Knowledge Retrieval In Table 4.
- How does HisTPT regularize the bounding box for object detection?
- What are the computational and storage overhead compared to TPT?

**Questions:**

See weakness.

**Limitations:**

Yes

---

> ### Author Rebuttal · Authors · 2024-08-06
>
> **[Response 1] Robustness of HisTPT to different confidence metrics:**
>
> Thank you for your suggestion! We conduct the suggested studies by adopting different confidence metrics, i.e., Softmax probability [a], MC dropout [b], and Mahalanobis distance [c], over Cityscapes semantic segmentation task with SEEM-Tiny. As shown in the table below, HisTPT can work with different confidence metrics. We chose entropy as it is simple and widely adopted.
>
> ||Softmax probability[a] | MC dropout[b] | Mahalanobis distance[c] | **Entropy (Default)**|
> |:----:|:----:|:----:|:----:|:---:|
> HisTPT| 44.6 | 44.7|44.5|**44.7**|
>
> [a] A Baseline for Detecting Misclassified and Out-of-Distribution Examples in Neural Networks, ICLR 2017.
>
> [b] Dropout as a Bayesian Approximation: Representing Model Uncertainty in Deep Learning, ICML 2016.
>
> [c] A Simple Unified Framework for Detecting Out-of-Distribution Samples and Adversarial Attacks, NeurIPS 2018.
>
> **[Response 2] Clarification of the ablation study design:**
>
> Thank you for pointing out this issue. For the ablation experiments without Adaptive Knowledge Retrieval, we compute the self-supervised loss $L_{self}$ with the regularized prediction $\hat p_n$ which is obtained without the adaptive weighting as in Eq.7.
>
> Specifically, for experiments using a single knowledge bank in Table 4, we compute $L_{self}$ by directly adopting the prediction of respective knowledge bank by Eq.6 as the final regularized prediction. For the rest experiments using more than a single knowledge bank, we compute $L_{self}$ by averaging the predictions of respective knowledge banks as the final regularized prediction.
>
> We will clarify this point in Section 4.4 in the updated manuscript.
>
> **[Response 3] How does HisTPT regularize boxes for object detection task:**
>
> We would clarify that, as discussed in Section Method in the main manuscript and Section Limitations in the appendix, HisTPT achieves effective text prompt tuning by regularizing the category prediction only, which is general and applicable to various vision foundation models on image classification, semantic segmentation and object detection.
>
> On the other hand, the bounding box prediction is the task-specific design for object detection task. We did not consider it during method design as we aim to build a general test-time prompt tuning framework that can work for various vision tasks.
> Nevertheless, we believe that introducing additional task-specific designs like regularizing bounding boxes for object detection tasks may improve the performance. We will investigate how to incorporate task-specific designs (e.g., regularizing bounding boxes) in our future work.
>
> **[Response 4] Computational and storage overhead compared to TPT:**
>
> Thank you for your suggestion! We conduct the suggested efficiency benchmarking with TPT in terms of run time and GPU memory usage, under the same setting with CLIP ResNet-50 on dataset Flowers-102 with a single GPU. As the table below shows, TPT has much longer inference time and higher GPU usage because 1) TPT involves heavy augmentations of test images (e.g., 64 augmentations for each test image) while HisTPT does not and 2) HisTPT efficiently memorizes the historical information by storing compacted data within fixed-size memory banks as described in Lines 155-157, 169-172 and 541-550.
>
>
> |Methods|Run time per image (s) | GPU memory usage (MB)|
> |:---:|:---:|:---:|
> |TPT | 0.66|4,210|
> |HisTPT| 0.12|3,888|

---

> ### Author Response · Authors · 2024-08-13
>
> Dear Reviewer 43p7,
>
> Thank you for your insightful feedback. We have carefully considered your questions and suggestions and have addressed them accordingly. We sincerely appreciate your constructive comments, which have helped strengthen our paper. As the discussion phase is nearing its conclusion, we would appreciate it if you could let us know if there are any additional questions or suggestions.
>
> Best regards,
>
> Authors

---

### Author Rebuttal · Authors · 2024-08-06

We would like to thank the reviewers for your insightful feedback and constructive comments. We are highly encouraged by the reviewers' acknowledgement that our proposed method has good scalability in various vision tasks [43p7,CdiG] and effectively explores memory in test-time prompt tuning [ejR5], the conducted experiments are comprehensive and solid [43p7,hT3z,ejR5,CdiG], and the paper is well-organized and easy to follow [43p7,hT3z].

We address the questions and concerns raised by each reviewer point-by-point in the respective threads below. In addition, attached below is the PDF containing the figures required for some responses.

---

### Decision · Program_Chairs · 2024-09-25

**Decision:**

Accept (poster)

**Comment:**

This paper proposes a method to improve test-time prompt tuning using historical information from memory bank in an online manner. In particular, the method introduces three types of knowledge banks: local, hard-sample, and global, aimed at memorizing learned knowledge to prevent performance degradation. The resulting method benefits from the design with more stable test-time-training performance over time. The method is applied to various tasks, including image classification, semantic segmentation, and object detection.

The reviews of the paper are borderline and slightly diverging. While the use of memory bank and the overall design does not stand out from the literature as ground breaking, the work does tackle an important problem in TPT and presents a very sound solution. The AC is also impressed that the work considered various visual recognition applications instead of focusing on just the image classification ones as other TPT works usually do. The AC thus considers this work with sufficient contributions and quality for publication.

That being said, some concerns from reviewers such as the discussion/comparison with other online/temporal/memory methods. It is also a pity that the authors did not highlight this method in video applications, even though there is one video dataset (UCF-101). It is not entirely clear to the readers how this work could benefit video applications.